# Counterfactual Explanations for 3D Point-Cloud Classifiers

## Abstract

Explainable AI (XAI) seeks to tackle the opacity of deep neural network decisions. Moving beyond the conventional focus on 2D imagery, our research constitutes the first method to provide Counterfactual Explanations (CEs) for 3D point cloud classifiers. Specifically, we introduce an approach to generate 3D CEs using a diffusion model whose goal is to maintain both semantic consistency and data fidelity in 3D contexts. To this end, we devise novel losses and constraints to boost the realism and practicality of counterfactual instances. Furthermore, we establish a new benchmark with evaluation metrics designed specifically for 3D point clouds, allowing for the assessment of future methods. Altogether, our contributions bridge a key gap in the field of explainability, steering towards more transparent and fair AI methodologies.

## 1 Introduction

Neural networks come with an inherent complexity that often obscures their decision-making process. This opacity is particularly problematic in critical fields such as healthcare, finance, and autonomous systems, where transparency, accountability, and fairness are imperative. Explainable AI (XAI) (Böhle et al.; Li et al., 2022) has emerged as a response to demystify neural networks' decision-making mechanisms.

Within XAI, Counterfactual Explanations (CEs) aim to provide insights through "what if" scenarios, generated by perturbing the sample input to the network. Unlike adversarial attacks, CEs prioritize semantic interpretability, leveraging techniques such as variational autoencoders (Rodríguez et al., 2021), GANs (Jacob et al., 2022) and diffusion models (Jeanneret et al.) to generate understandable input modifications that alter the model's output. While rapid progress has been achieved in the development of CEs for image-processing networks, to the best of knowledge, no such effort has yet been made for 3D processing ones.

Our work therefore diverges from existing CE methods by not focusing on 2D image-based models but embracing 3D data processing challenges, with a particular emphasis on point cloud classifiers. This direction seeks to address the void in counterfactual generation for 3D perception models, introducing new dimensions to model debugging and comprehension.

Our exploratory effort begins with the perturbation of 3D point clouds to challenge classifiers that interpret 3D data directly. Drawing inspiration from recent advancements in image counterfactuals (Jeanneret et al.; 2023), which leverage the ability of diffusion models to produce high-quality images for counterfactual generation, we devise a method for generating counterfactuals in 3D point clouds using diffusion (Zeng et al., 2022). However, in contrast to the methods of (Jeanneret et al.) and (Jeanneret et al., 2023), in our approach, the diffusion model is integrated into counterfactual optimization through a score distillation loss (Poole et al., 2022). This technique eliminates the necessity to corrupt the original sample at the start of the counterfactual generation process, thereby preserving information from the original sample. Furthermore, it enhances efficiency by requiring only a single diffusion step per optimization step and alleviates the need for back-propagation through the diffusion model.

We conduct a thorough analysis of both the quantitative and qualitative effectiveness of our approach to generating CEs for point cloud classifiers (Wang et al., 2019; Pang et al.), optimizing their performance against criteria specifically tailored to the structure of point-cloud data. To facilitate this

analysis, we introduce metrics designed to evaluate different aspects of the counterfactual examples, such as their validity, proximity, diversity, realism, and sparsity (Jeanneret et al., 2023). We compare the performance of this diffusion based pipeline with not only an adversarial attack that affects the point clouds structure directly, but also with an Adversarial Variational AutoEncoder (AAE) baseline inspired by (Zamorski et al., 2019). This choice of baselines is motivated by the lack of previous CE work in the 3D domain.

Our contributions can thus be summarized as follows:

- We are the first to tackle the problem of CE generation in the 3D domain.
- We introduce a diffusion-based method and a tailored loss for generating counterfactual explanations applicable to 3D point cloud classifiers.
- We are the first to study CEs for multi-class classifiers; even in the image domain, previous studies focused on binary cases only.
- We formulate specialized evaluation metrics designed for assessing 3D CEs, and provide a benchmark for 3D CE generation for point cloud classifiers.

We will make our code publicly available.

## 2 RELATED WORK

### 2.1 EXPLAINABLE AI

Explainability in AI can be classified into two primary categories: Designing ad-hoc techniques via machine learning architectures that are inherently interpretable, as advocated for by recent studies (Bohle et al.; Böhle et al.; Huang & Li; Rymarczyk et al.); and developing post-hoc techniques, which are designed to scrutinize pre-existing models and their predictions either from a global (Tan; Li et al., 2022) or local perspective (Ribeiro et al., 2016; Chattopadhay et al.). Global methods elucidate the overall functioning of the model, whereas local methods delve into the model's decision-making process for individual instances, explaining the rationale behind specific decisions. Our research is dedicated to local post-hoc explanations.

In this context, many solutions have been proposed to provide local explanations for 2D image-based models (Li et al., 2022). These techniques primarily utilize saliency maps (Chattopadhay et al.; Jalwana et al.; Lee et al.; Kim et al., b), local approximations (Ribeiro et al., 2016; Guidotti et al., 2018), concept attributions (Kim et al., a; Ghorbani et al.; Kolek et al.), and prototypes or critiques (Kim et al., 2016; Gurumoorthy & Dhurandhar, 2018). In particular, our research focuses on counterfactual explanations, another significant branch of local post-hoc explanation strategies, which we thus discuss in more detail below.

### 2.2 COUNTERFACTUAL EXPLAINATIONS AND FAILURE MODES

CEs serve as a critical tool in understanding the decision-making processes of AI models by addressing the "what would have happened if" question. More precisely, CEs seek to identify an alternative sample, $\tilde{x}$, that is minimally different from an original input $x$, yet leads to a different outcome from the model. The purpose of counterfactual explanations lies in their ability to produce modifications that are understandable and meaningful to humans, thereby elucidating how a model arrives at its conclusions. This represents a critical distinction between counterfactual explanations and adversarial attacks; the latter typically involve modifications that are not interpretable by humans. The importance of including an interpretability constraint to generate such explanations has been widely acknowledged, with various strategies proposed to this end (Van Looveren & Klaise, 2020; Thiagarajan et al.; Hvilshøj et al.; Boreiko et al.). In the case of image classifiers, the standard constraint for semantic changes in input images involves using generative tools to force the counterfactual sample to remain in the image manifold (Rodríguez et al., 2021; Jacob et al., 2022; Zemni et al., 2022; Jeanneret et al., 2023). Image-based generative methods struggle with semantic control (Zemni et al., 2022; Jeanneret et al., 2023). Leclerc et al. (2021) introduces a method to debug image-based deep learning models using 3D rendering, focusing on 3D setting manipulation for interpretability. The method's efficacy depends more on the rendering software than the framework.

The realism evaluation of failure modes is challenging due to the limitations in Leclerc et al. (2021), which do not fully test the renderer's capabilities. As rendering systems are a limitation for the realism of a CE and image-based CE generation methods lack of controls in the parameter space, we offer to extend the frontier of counterfactual reasoning into the 3D realm by first raising the question of the possibility to develop CEs for 3D data structures. Our research translates 2D image-based CE methodologies to 3D point cloud classifiers, aiming to broaden CE applications across data types and enhance explainable AI insights. Following Rodríguez et al. (2021), we utilize unconstrained variational autoencoders for generating semantically meaningful CEs, optimizing through a model's latent space (Cohen et al., 2021; Rodríguez et al., 2021) with gradient descent (Joshi et al., 2019; Rodríguez et al., 2021). Moreover, (Jeanneret et al.; 2023) demonstrate diffusion-based generative models for CE, guiding our analysis of such models in 3D contexts.

## 2.3 POINT CLOUD CLASSIFICATION

Point clouds, as a shape representation method, offer inherent simplicity and efficiency. These scattered collections of points, outputted by advanced 3D sensing technologies like LiDAR scanners, enable us to streamline our workflow—saving computational resources and reducing processing time. Additionally, their flexibility and lightweight nature make point clouds suitable for diverse applications across computer graphics, vision, robotics, and virtual reality.

Point cloud classifiers which process directly point clouds aggregate either local features (Qi et al., b; Qian et al.) or global ones (Qi et al., a). The significance of local geometry in 3D recognition tasks is proved by Wang et al. (2019) and its introduction to a new convolutional paradigm and a dynamic graph CNN (DGCNN) which captures of local geometric features in point clouds and ensures their arrangement remains invariant. Pang et al. developed Point-MAE, a masked autoencoder approach for the self-supervised learning of point clouds.

## 2.4 POINT CLOUD GENERATION

To develop CEs for 3D point cloud classifiers, we will exploit methods for generating point clouds. Recent years have witnessed a surge of interest in developing generative models specifically tailored for 3D point clouds, employing various strategies such as autoencoders (Zamorski et al., 2019), flow-based models (Pumarola et al., 2019), and energy-based models (Luo & Hu, 2021). These sophisticated generative approaches offer a versatile toolkit for the modeling of complex data types, particularly for the intricate details encapsulated in 3D objects and point clouds.

In particular, the Adversarial Autoencoder (AAE) (Zamorski et al., 2019) has emerged as a suitable baseline for comparison with our method. It simultaneously learns a representation in latent space and generates 3D shapes from this learned space, innovating beyond the Variational Autoencoder by integrating adversarial training for enhanced 3D data processing and output.

Furthermore, the domain of 3D point cloud generation has been profoundly influenced by the advent of diffusion models (Ho et al., 2020; Zhou et al.; Luo & Hu), representing the forefront of deep learning research. Among these, the LION model (Zeng et al., 2022) encodes the complex distributions of 3D shapes through latent space representations, thus facilitating the creation of highly detailed and varied 3D point clouds. As will be discussed below, we investigate the use of LION to produce CEs for 3D point cloud classifiers.

## 3 METHOD

### 3.1 ENCODE-DECODER PIPELINE

Our pipeline for generating 3D counterfactual explanations is illustrated in Fig. 1. Unlike adversarial attacks that directly manipulate the point cloud, we perform modifications within a learned latent space. Starting from a sample $\mathbf{x} \in \mathbb{R}^{2048 \times 3}$, we use an encoder $E(\cdot)$ to obtain a latent representation $\mathbf{z} = E(\mathbf{x})$. This representation is the starting point of the optimization process for the generation of a counterfactual sample. We then add a learnable residual vector $\boldsymbol{\delta}$ to $\mathbf{z}$ so as to generate a counterfactual sample $\tilde{\mathbf{x}} = D(\mathbf{z} + \boldsymbol{\delta})$ using the decoder network $D(\cdot)$. The modified sample $\tilde{\mathbf{x}}$ is then given as input to the classifier $f$. The classifier is predefined, and, in our experiments, we focus

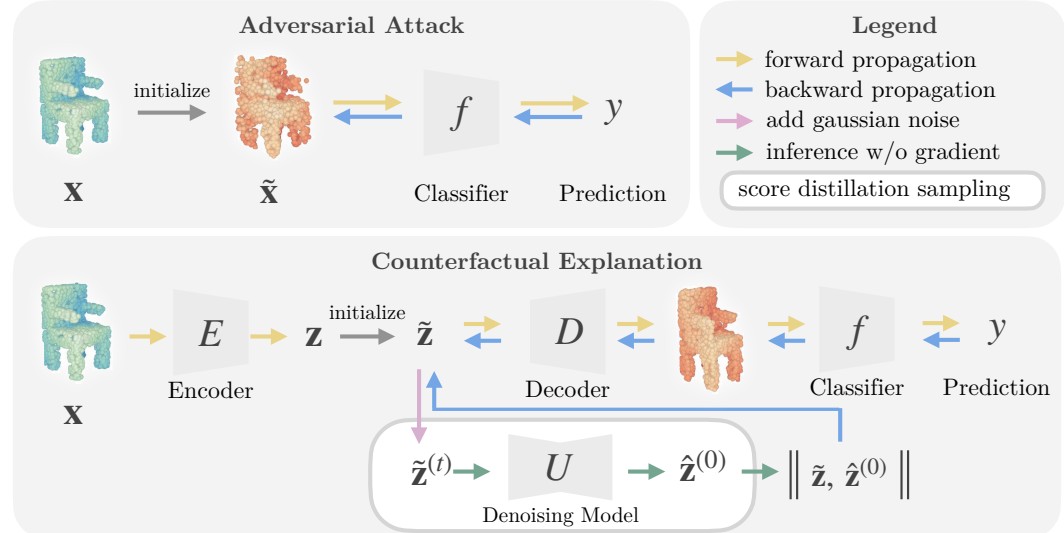

Figure 1: **Counterfactual Explanation Pipeline**. In this figure we contrast the pipeline for Adversarial Attack with our diffusion-based counterfactual explanation method.

on binary classification tasks on point cloud data. Generating a CE is thus achieved by optimizing $\boldsymbol{\delta}$ such that $f(\tilde{x}) \neq f(x)$, yet, in contrast with adversarial attacks, $\tilde{x}$ should represent a semantically-meaningful change over the original input $\mathbf{x}$. In practice, we simultaneously optimize $K$ of these perturbations in parallel for a single input sample, with each one starting from a small random perturbation $\boldsymbol{\delta}^i \sim \mathcal{N}(\mathbf{0}, 0.01 \cdot \mathbf{I})$.

### 3.2 3D CEs via Diffusion Score Distillation

Diffusion models have established themselves as efficient generators (Dhariwal & Nichol, 2021). Their use in providing counterfactual explanations in the 2D domain has enhanced the realism of the generated samples (Jeanneret et al.; 2023). We make use of the LION point-cloud diffusion framework (Zeng et al., 2022). This framework is based on the PVCNN (Liu et al., 2019) autoencoder, which incorporates an encoder $E(\cdot)$ and a decoder $D(\mathbf{z}_g, \mathbf{z}_l)$ acting on two latent vectors: A global latent vector $\mathbf{z}_g \in \mathbb{R}^{128}$, which is similar to the latent space in typical autoencoders, and a local latent vector $\mathbf{z}_l \in \mathbb{R}^{8192}$, which encodes information at a finer granularity. Notably, the local latent space is not meaningful everywhere; a denoising model $U(\mathbf{z}_l^{(t)} | \mathbf{z}_g, t), t \in [1, 500] \cap \mathbb{N}$ is introduced to recover a meaningful local latent code from a noisy one. Specifically, the network $U$ estimates the noise in the current local latent code, conditioned on the global latent code and the current time-step $t$.

To search for CEs, we therefore optimize two vectors $\boldsymbol{\delta}_g$ and $\boldsymbol{\delta}_l$, corresponding to the global and local latent spaces, respectively. We introduce for the first time the score distillation loss (Poole et al., 2022) into the search for counterfactual explanations. It is expressed as

$$L_{\text{sds}} = \frac{1}{2}\mathbb{E}_{t \sim \mathcal{U}(1,T)}\left[w(t)\left\|\hat{\mathbf{z}}_l^{(0)} - \mathbf{z}_l\right\|^2\right], \quad \hat{\mathbf{z}}_l^{(0)} = \frac{1}{\sqrt{\bar{\alpha}_t}}(\mathbf{z}_l^{(t)} - \sqrt{1 - \bar{\alpha}_t}\, U(\mathbf{z}_l^{(t)}|\mathbf{z}_g, t)), \quad (1)$$

where $\mathbf{z}_l^{(t)} = \sqrt{\bar{\alpha}_t}\mathbf{z}_l + \sqrt{1 - \bar{\alpha}_t}\boldsymbol{\epsilon}$ is the corrupted version of $\mathbf{z}_l$; $t \sim \mathcal{U}\{75, 425\}$ is chosen randomly; $\bar{\alpha}_t$ is defined by the denoising schedule; $\boldsymbol{\epsilon} \sim \mathcal{N}(\mathbf{0}, \mathbf{I})$ is a random Gaussian noise; $\hat{\mathbf{z}}^{(0)}$ is the denoising endpoint predicted by a denoising model $U$; $w(t)$ is a weighing function that depends on the time-step $t$. As mentioned in (Poole et al., 2022), SDS optimization is robust to the choice of $w(t)$, and we use $w(t) = 1$.

This objective has the advantage over earlier diffusion-based methods (Jeanneret et al., 2023; Jeanneret et al.) of eliminating the necessity to back-propagate through multiple chained steps of denoising, requiring only one gradient-free forward step, thereby greatly reducing computation costs.

## 3.3 TRAINING LOSS TERMS

We provide additional constraints during the optimization process so that the complete loss term consists of four more terms. The first one is the counterfactual loss $L_{cf}$, which aims to deceive the point cloud classifier. Considering a binary classifier, let $y \in {0, 1}$ be the label corresponding to an input sample $\mathbf{x}$. The counterfactual loss is defined as the confused binary cross entropy, i.e.,

$$L_{cf} = -(1 - y) \log f(\tilde{\mathbf{x}}) - y \log(1 - f(\tilde{\mathbf{x}})) . \tag{2}$$

The second loss is the proximity loss $L_{prox}$, defined as an $L1$ regularizer on the edited point cloud $\tilde{\mathbf{x}}$ and the latent space perturbation $\boldsymbol{\delta}$. Denoting the $i^{th}$ channel of the latent vector by $\boldsymbol{\delta}_i$ and the $j^{th}$ point in the point cloud by $\mathbf{x}_j$, the proximity loss is expressed as

$$L_{prox} = \sum_i \|\boldsymbol{\delta}_i\|_1 + \gamma \sum_j \min_k \|\mathbf{x}_j - \tilde{\mathbf{x}}_k\|_1 , \tag{3}$$

where $\gamma$ is a scalar weight parameter. The third one is a regularizer to encourage the generated counterfactual explanation to retain a reasonable point cloud structure. To this end, we constrain the distance from a point to its $K$-nearest-neighbors. Denoting the distance from the $j^{th}$ point to its $k^{th}$ nearest neighbor in the point cloud by $d_{j,k}$, the loss is formulated as a negative log likelihood based on a $K$-nearest-neighbors distribution prior

$$p(d_{j,k}|\rho, k) = \frac{2(\pi\rho)^k d_{j,k}^{2k-1}}{(k-1)!} \exp(-\rho\pi d_{j,k}^2). \tag{4}$$

This prior assumes that the points in a point cloud are randomly uniformly distributed on the object surface. In this case, the nearest-neighbor distribution is essentially the two dimensional version of an exponential distribution, where $\rho$ represents the density of the points, the derivation of this distribution is presented in Sec. A.1. Omitting the terms without gradients, we obtain

$$L_{st} = \sum_{j,k} -\log p(d_{j,k}|\rho, k) = \sum_{j,k} \pi\rho d_{j,k}^2 - 2k \log d_{j,k} . \tag{5}$$

The derivatives of this loss with respect to the distance $\frac{\partial}{\partial d_{j,k}} L_{st} = 2\pi\rho d_{j,k} - \frac{2k}{d_{j,k}}$ contains a linear term and a reciprocal term, penalizing points that are either too far from each other or too close.

The last loss term is a diversity loss to encourage diversified counterfactual explanations. It is expressed as

$$L_{div} = \sum_i \sum_{j>i} \frac{1}{1 - \theta(\boldsymbol{\delta}^i, \boldsymbol{\delta}^j)} , \tag{6}$$

where $\theta(\boldsymbol{\delta}^i, \boldsymbol{\delta}^j)$ denotes the cosine similarity between perturbation vectors. The formulation of this loss is based on the potential energy within an imaginary system characterized by an inverse-squared repulsive force acting between every pair of latent points.

Altogether, the optimization objective is thus specified as

$$L_{Diff} = L_{cf} + \lambda_{sds} L_{sds} + \lambda_{prox} L_{prox} + \lambda_{st} L_{st} + \lambda_{div} L_{div} . \tag{7}$$

## 4 EXPERIMENTS

**Baselines.** Our experiments assess quantitatively and qualitatively the performance of our diffusion-based CE generation method by comparing it against three baselines. Specifically, we compare our CE generation method with an *Adversarial Attack* (AA) baseline (Madry et al., 2018). Given an input sample $\mathbf{x} \in \mathbb{R}^{2048 \times 3}$ and the classifier $f : \mathbb{R}^{2048 \times 3} \rightarrow \{0, 1\}$, we compute a displacement $\boldsymbol{\delta} \in \mathbb{R}^{2048 \times 3}$ such that $\mathbf{x} + \boldsymbol{\delta} = \tilde{\mathbf{x}}$ and $f(\tilde{\mathbf{x}}) \neq f(\mathbf{x})$. The computation of such a displacement is done by optimizing the counterfactual loss $L_{cf}$. We also study a constrained version of the adversarial attack, namely *Adversarial Attack +* (AA+), which combines Eq. 2, Eq. 5 and the last term in Eq. 3, i.e.,

$$L_{Adv+} = L_{cf} + \lambda_{st} L_{st} + \lambda_{prox} \sum_i \min_j \|\mathbf{x}_i - \tilde{\mathbf{x}}_j\|_1 . \tag{8}$$

Following Section 3, we also compare our *Diffusion* based method with an *AutoEncoder* (AE) based method. This leverages another encoder-decoder model where the latent space is defined by a pre-trained AAE (Zamorski et al., 2019). The optimization objective for AE is specified as

$$L_{AE} = L_{cf} + \lambda_{prox} L_{prox} + \lambda_{st} L_{st} + \lambda_{div} L_{div} . \tag{9}$$

**Classifiers.**   We evaluate these methods on the DGCNN (Wang et al., 2019) architecture, as elaborated in Sec. 2.Following common practice (Jacob et al., 2022; Zemni et al., 2022; Jeanneret et al.; 2023; Rodríguez et al., 2021), our investigation firstly centers on binary tasks. In the context of point cloud classification tasks, we train the model on different object categories available in the ShapeNetCore dataset. Considering the class $c \in$ [chair, airplane], the classifiers have been trained to classify a point cloud to belong to $c$ or not. We also conduct experiments using a DGCNN model trained to classify point clouds into 55 ShapeNet classes (Multi-Class). During the CE generation process, we specify a target class, which is selected as the class with the second-highest probability score from the model's classification output. We augment the training dataset to train them by sampling at each epoch 2048 points randomly among the 15k points available for each sample.

**Datasets.**   We employ the ShapeNet dataset (Chang et al., 2015) which is a large repository of 3D CAD models. ShapeNetCore is a subset of ShapeNet that includes nearly $51,300$ unique 3D models. It provides 55 common object categories and annotations. We conduct the same set of experiments on the two object categories with the largest number of samples *chairs* and *airplanes* from the ShapeNetCore dataset, containing $6,590$ and $4,045$ samples, respectively.

## 4.1   Evaluation Metrics

We employ evaluation criteria that assess different aspects of the counterfactual explanations, including their validity, diversity, sparsity or proximity, transition probabilities, and the realism of the generated counterfactual images (Jeanneret et al., 2023).

**Validity.**   Following common practice (Jeanneret et al., 2023), we start by measuring the flip rate (FR), indicating the proportion of cases where the algorithm finds at least one counterfactual explanation. Our studies show that many counterfactual explanations are trivial, identified by the generation of outliers—points far from the majority. We thus define non-trivial counterfactuals as meeting specific structural criteria, labeling outliers based on their distance from the nearest neighbor within the first or last permille of the distribution. Considering this definition, we report the non-trivial flip rate (NTFR) and the mean number of outliers (MNO).

**Proximity.**   Modifications within point clouds are measured using the Chamfer distance (CD), and perceptual changes are quantified with LPIPS (Zhang et al., 2018). We assess the perceptual distance between the original samples and the counterfactuals using the pretrained SimpleView (Goyal et al., 2021), which projects point clouds into 2D from various angles for 2D CNN classification. This approach, mimicking human perception, bridges the 3D and 2D realms. SimpleView, trained on the entire ShapeNetCore dataset (Chang et al., 2015), serves as an oracle for our perceptual metric.

**Sparsity.**   We evaluate sparsity with a revised version of the mean number of attributes changes (MNAC) (Rodríguez et al., 2021; Jacob et al., 2022; Jeanneret et al., 2023). Considering a predefined set of attributes that defines semantic parameters in the data manifold, we expect our CE generation method to change a minimum number of them for each generation process.

Let the SimpleView encoder be defined by $\Omega : \mathbb{R}^{2048 \times 3} \to \mathbb{R}^{128}$. Let $\mathbf{x}, \tilde{\mathbf{x}} \in \mathbb{R}^{2048 \times 3}$ be respectively an original point cloud sample and one of its counterfactual examples. Considering the data distribution $X$, we assume that for each channel $i \in [1, 128] \cap \mathbb{N}$ of the encoded space, we have $\Omega(X)_i \sim \mathcal{N}(\mu_i, \sigma_i)$. We then compute the number of attributes changes as

$$\text{NAC} = \sum_{i=1}^{128} \frac{\|\mathbf{\Gamma}\|_1}{128} \frac{c(i)}{\sum_{j=1}^{128} |\mathbf{\Gamma}_{ij}|}, \quad c(i) \coloneqq \mathbb{1} \left\{ \left| \text{CDF}\left( \frac{\Omega(\mathbf{x})_i - \mu_i}{\sigma_i} \right) - \text{CDF}\left( \frac{\Omega(\tilde{\mathbf{x}})_i - \mu_i}{\sigma_i} \right) \right| \geq 0.5 \right\},$$
$$\tag{10}$$

where $\mathbf{\Gamma}$ denotes the correlation matrix, calculated from the training dataset, and $c(i)$ serves as a robust indicator of attribute change in the $i^{\text{th}}$ channel, with CDF denoting the cumulative distribution

| Classifier | Method | FID ↓ | FR ↑ | NTFR ↑ | MNO ↓ | CD ↓ | LPIPS ↓ | MNAC ↓ | Div. ↑ |
|---|---|---|---|---|---|---|---|---|---|
| Binary *chair* vs. *not chair* | Adv. Attack (AA) | 1.448 | 0.966 | 0.200 | 46.630 | 4.401 | 0.874 | 22.897 | 0.725 |
| | Adv. Attack + (AA+) | 0.624 | 0.958 | 0.238 | 29.933 | 2.299 | 0.585 | 10.609 | 0.538 |
| | AutoEncoder (AE) | 2.019 | 0.964 | 0.600 | 37.786 | 20.130 | 1.717 | 17.491 | 1.658 |
| | Diffusion (Diff) | 0.154 | 0.936 | 0.850 | 8.585 | 10.364 | 1.003 | 7.243 | 0.670 |
| Binary *airplane* vs. *not airplane* | Adv. Attack (AA) | 0.813 | 1.000 | 0.205 | 20.731 | 3.428 | 0.626 | 23.241 | 0.429 |
| | Adv. Attack + (AA+) | 0.343 | 1.000 | 0.000 | 72.334 | 2.084 | 0.430 | 7.877 | 0.291 |
| | AutoEncoder (AE) | 1.316 | 0.965 | 0.010 | 66.360 | 19.795 | 1.193 | 53.766 | 0.990 |
| | Diffusion (Diff) | 0.673 | 0.926 | 0.864 | 9.393 | 17.443 | 1.015 | 30.658 | 0.501 |
| Multi-Class *chair* vs. *others* | Adv. Attack (AA) | - | 0.094 | 0.052 | 25.342 | 2.069 | 0.034 | 1.276 | 0.025 |
| | Adv. Attack + (AA+) | - | 0.084 | 0.06 | 13.862 | 1.395 | 0.023 | 0.194 | 0.018 |
| | Diffusion (Diff$_{chair}$) | 0.844 | 0.440 | 0.270 | 15.596 | 13.056 | 1.082 | 8.922 | 0.751 |
| | Diffusion (Diff$_{all}$) | 1.106 | 0.615 | 0.425 | 11.958 | 20.043 | 1.218 | 12.258 | 0.790 |

Table 1: **Quantitative comparison** on attacking binary and multi-class classifiers. In the Multi-Class setting, we focus on inputs from the chair category. Diff is either trained on the whole ShapeNet dataset, Diff$_{all}$, or on the chair category only , Diff$_{chair}$.

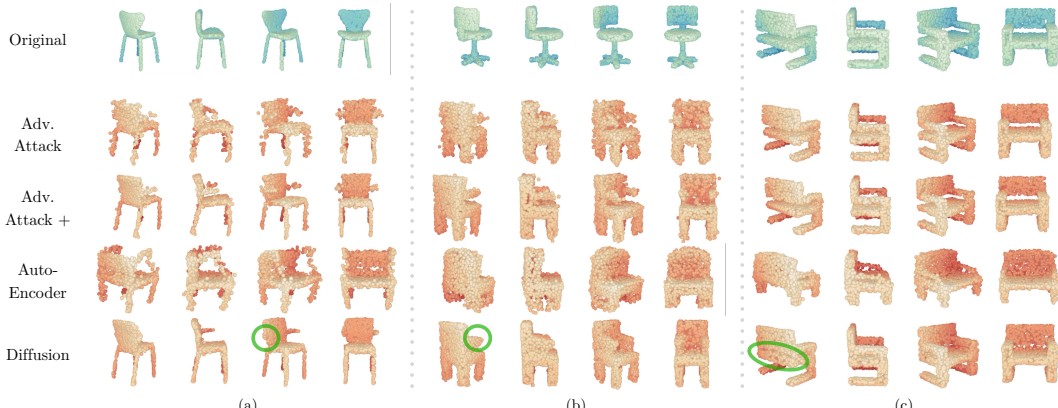

Figure 2: **Qualitative Comparison** of our pipelines for the *chair* category in ShapeNetCore. Semantical changes are highlighted with green circles.

function of $\mathcal{N}(0, 1)$. The channel-wise statistics $\{\mu_i\}$ and $\{\sigma_i\}$ are calculated from the training dataset. This method measures any important change in a channel with respect to its distribution. It accounts for potential entanglement in different channels via the normalization of the count of attribute changes by the sum of their correlation coefficients. The reported MNAC is obtained by averaging over all discovered CEs.

**Diversity.** Our diversity metric (*Div*) is inspired by (Jeanneret et al., 2023) and is quantified as the average pairwise perceptual distance between different counterfactuals optimized for the same input. We also use the SimpleView (Goyal et al., 2021) based LPIPS as a perceptual metric for diversity.

**Realism.** To assess realism, we use the Frechet Inception Distance (FID) (Heusel et al., 2017), comparing original samples with their counterfactuals (Heusel et al., 2017; Jeanneret et al.; 2023; Augustin et al., 2022). However, Jeanneret et al. (2023) highlight FID's bias, as unchanged parts in the data skew the results. They propose sFID, randomly dividing the dataset, creating CEs, and comparing them across subsets, averaged over ten iterations. Given the FID's sensitivity to sample size (Barratt & Sharma, 2018) and the variability in CE numbers, we use a different approach. We randomly sample 2,000 real and 500 generated samples, calculating the mean FID. Real samples come from the training split, and generated CEs from the test set, ensuring an unbiased metric.

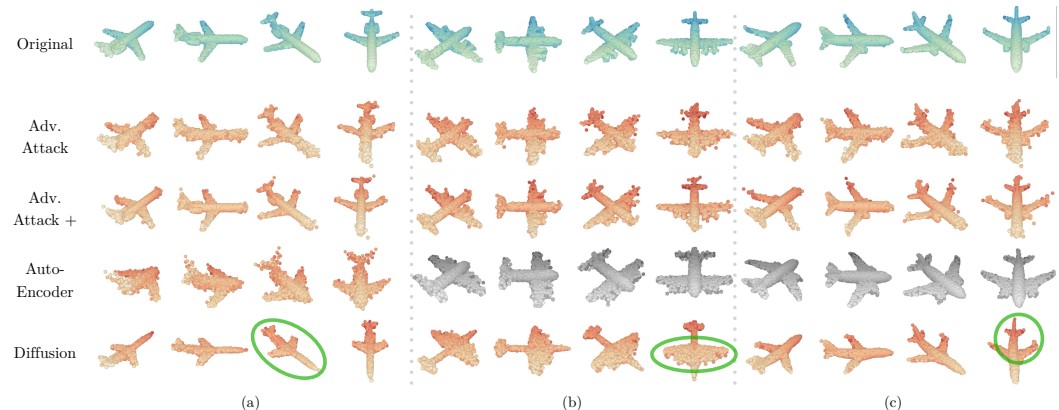

Figure 3: **Qualitative Comparison** of our pipelines for the *airplane* category in ShapeNetCore. Semantical changes are highlighted with green circles. Black-and-white denote samples which fail to decieve the classifier.

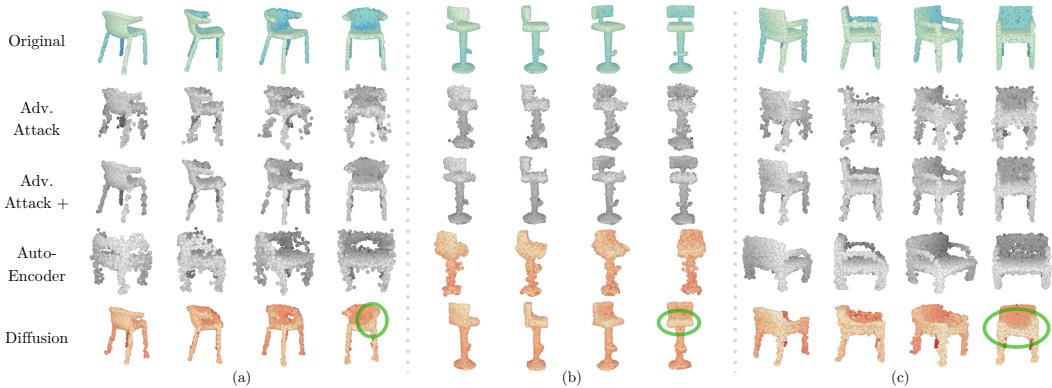

Figure 4: **Qualitative Comparison** of our pipelines for attacking a multi-class classifier. Semantical changes are highlighted with green circles. Black-and-white denote samples which fail to decieve the classifier.

## 4.2 RESULTS

### 4.2.1 QUANTITATIVE PERFORMANCE

In our comparative analysis shown in Table 1, Diffusion (*Diff*) emerges as the superior technique in terms of FID, NTFR, MNO. This performance indicates that the CEs generated by diffusion method not only closely resemble the original instances within ShapeNetCore's chair subset but also effectively identify and manipulate the visual features critical for the predictions made by the target classifier. Notably, despite yielding relatively high Chamfer Distance and LPIPS values, the MNAC remains lower than the AutoEncoder (AE) method. This indicates that the edits introduced are sparse, which is desirable for generating counterfactuals, as it facilitates the discovery of spurious correlations. Altogether, these results show the ability of our method to produce realistic and pertinent CEs. In the Multi-Class setting, both $\text{Diff}_{all}$ and $\text{Diff}_{chair}$ outperform (FR, NTFR, MNO) the adversarial attack methods which cannot provide enough counterfactual samples to measure the FID as the flip ratio is very low. The higher Chamfer Distance, LPIPS, MNAC and DIV for the diffusion methods is expected as we allow, again, structural deformations to get semantic modification on the samples. In comparison of the two diffusion method in that Multi-Class settings, we notice that the diffusion method trained on a single category of ShapeNet is restrained in the modifications it can apply to a sample, which explains the lower FID, CD, LPIPS, MNAC and DIV. However, the diffusion method trained on the whole dataset provides better flip ratios and diversity as the latent space a denser data space and allows further modification on the structure of the sample.

The *AE* baseline is distinguished by its high flip ratio. It provides a certain degree of flexibility in modifying the input sample, yielding to more diverse CEs than those generated by other techniques. Nonetheless, this advantage is tempered by a notable drawback: The CEs produced by the autoencoder exhibit a reduced resemblance to the original samples. This reduction in similarity is attributed to the inherent constraints of autoencoders in data representation, particularly when contrasted with the more nuanced capabilities of diffusion models. Such constraints may hinder the exploration of failure modes and the detection of spurious correlations, as evidenced by the elevated MNAC score. This suggests an excessive alteration of features for the same sample, undermining the fidelity of the CEs. In contrast with *Diff*, the *AE* latent space is only 2048 dimension wide, which makes it hard for us to apply local changes to the point cloud sample.

Finally, we incorporate *AA* and *AA+* as a reference, despite it not being a primary focus of our work. These methods outperforms the others in terms of the Chamfer Distance and LPIPS. The adversarial attack's core strategy, which minimizes the perturbation applied to the input sample, thereby maintains the original structure as much as possible. Such minimal disturbance contrasts with the intention behind CEs, which aim for changes that are not only perceptible but also interpretable by humans.

However, despite having a high flip ratio when attacking the binary classifiers, the results of adversarial attack contain a large number of outliers, resulting in a very low non-trivial flip ratio. Besides, these methods fails to attack the multi-class classifiers.

The improvements from AA to AA+ demonstrate the effectiveness of our regularization terms. However, these losses are not sufficient to make the method produce semantically meaningful changes. These attributes align with the qualitative observations, shown in the qualitative results in Fig. 2, and underscores the critical role of visual evaluation in comprehensively understanding the efficacy of our CE strategy tailored for point cloud classifiers.

### 4.2.2 QUALITATIVE OBSERVATIONS

Examining the qualitative differences between the adversarial attack and the two CE pipelines, a clear visual distinction in the deformations applied to the chairs emerges, aimed at altering the classifier's prediction. *AA* yields outputs that, while visually closer to the original sample as shown in Fig. 2c, exhibit a compromised structure with a significant number of outliers in comparison to Fig. 2a and Fig. 2b. *AA+* exhibits an alleviation this problem and reduces the number of outliers, such as shown in Fig. 2a and Fig. 2b, but still fail to produce semantically meaningful changes.

The *Diff* pipeline provides a way to visually assess the importance of certain features in point cloud classifier predictions. Through Fig. 2 and 4, *Diff* effectively isolates and modifies specific chair attributes, influencing the classifier predictions. Notable modifications, such as modification of a single armrest and yielding an asymmetric structure are observed in Fig. 2a, and Fig. 4a,c. Similarly, for airplanes, the *Diff.* pipeline is observed to change the classifier's output by, for example, bending the body of the airplane (Fig. 3c). Overall, Fig. 3 shows that *Diff* tends to remain geometrically close to the origin input which supports the quantitative results previously shown. By contrast, *AE* leads to more pronounced visual changes in the sample's features, aligning with its quantitative performance. Remarkably, *AE* tends to increase the dimensions of both the backrest and seat while reducing the size of the legs. Additionally, our CE pipelines demonstrate the capability to unveil failure patterns. A recurring pattern in results from out *Diff.* method involves breaking the symmetry in the input sample, suggesting that the model might have a tendency towards relying on symmetric structures.

### 4.3 ABLATION STUDY

We conduct an ablation study with the diffusion based model, and show the metrics evaluated on the DGCNN classifier trained on chairs in Table 2. Qualitative samples are shown in Fig. 5 The other results are shown in the supplementary material. In this study, we start from the diffusion model with only the counterfactual loss enabled, gradually adding loss terms to the algorithm and evaluating the results. Our results show that the score distillation loss $L_{sds}$ significantly reduces the FID, while boosting the flip rate and especially the non-trivial flip rate and MNO at the same time. It also brought improvement to the diversity of counterfactual explanations, possibly due to the stochasticity in the process of adding noise and denoising. The proximity loss $L_{prox}$, as expected, reduces the Chamfer distance and LPIPS significantly. The structure loss $L_{st}$ boosts the non-trivial

| | FID ↓ | FR ↑ | NTFR ↑ | MNO ↓ | CD ↓ | LPIPS ↓ | MNAC ↓ | Div. ↑ |
|---|---|---|---|---|---|---|---|---|
| $L_{\text{cf}}$ | 0.836 | 0.958 | 0.506 | 22.453 | 4.153 | 0.811 | 15.340 | 0.661 |
| $+L_{\text{prox}}$ | 0.368 | 0.960 | 0.682 | 15.764 | 3.527 | **0.649** | **6.034** | 0.482 |
| $+L_{\text{div}}$ | 0.408 | 0.960 | 0.656 | 17.104 | 4.045 | 0.673 | 6.894 | 0.579 |
| $+L_{\text{st}}$ | 0.401 | **0.962** | 0.722 | 15.311 | **3.429** | **0.649** | 6.847 | 0.561 |
| $+L_{\text{sds}}$ | **0.154** | 0.936 | **0.850** | **8.585** | 10.364 | 1.003 | 7.243 | **0.670** |

Table 2: **Ablation study** on attacking DGCNN trained on chairs with our diffusion-based method.

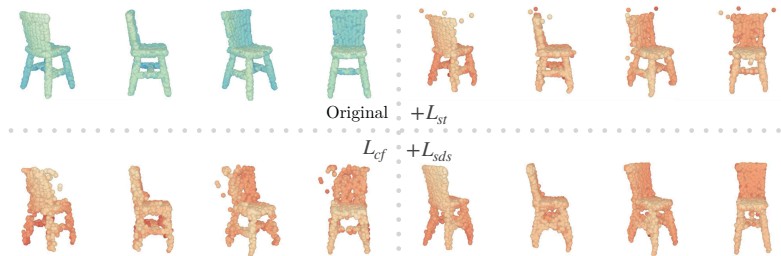

Figure 5: **Qualitative ablation study** on attacking DGCNN trained on chairs with our method.

flip rate while reducing the MNO. The diversity loss, apart from improving the diversity metric, benefits FID, MNO and MNAC as well. The diversity loss $L_{\text{div}}$ boosts diversity as expected while improving FID at the same time.

## 5 CONCLUSION

Our study marks an advancement in the field of XAI by introducing pioneering methods for generating CEs for 3D data structures, starting here with point clouds. By leveraging diffusion models, we have established a novel approach that ensures semantic consistency and high data fidelity within 3D contexts. Our development of new losses and constraints has substantially enhanced the realism and practicality of our counterfactual instances.

The diffusion based method and its objective developed for the counterfactual examples generation has been developed to improve the memory footprint and the computational costs of diffusion based CE generation.

Moreover, the introduction of specifically designed evaluation metrics for 3D point clouds sets a new benchmark, facilitating future research and validation of methods in this area.

Our method, however, has been evaluated on a for a limited amount of tasks and was thus constrained on a certain dataset. Although this follows the standard practice in image CEs, assessing it for other deep learning tasks and classifiers would strengthen its validation. It would be also interesting for future research to consider generative methods that are not constrained to the point cloud domain in order to attack diverse tasks and 3D structures (Loper et al., 2015).

Furthermore, the control we have on the CE generation is limited. Although the diffusion based CE generator provides local modifications on the samples or CEs that are visually similar to the input, we need a human analysis as a post-process to understand the results and which semantic properties of the sample the classifier is sensitive to. Thus exploring generative methods with more control on the part of the object to modify would ease the interpretation of the results (Hui et al., 2022).

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

# A    APPENDIX

In this appendix, we present the derivation of the K-nearest-neighbors distribution prior which we use to derive our structure loss $L_{st}$ (Sec. A.1) and implementation details (Sec. A.2).

## A.1    NAREST-NEIGHBOR DISTRIBUTION

Assuming points are randomly uniformly distributed on a 2D plane, let $\rho$ be the density of points per unit area, then the expected number of points within a circle of radius $r$ is $\rho\pi r^2$. The number of points $N(r)$ within a radius $r$ follows a Poisson distribution with mean $\mu = \rho\pi r^2$.

The probability that the distance to the $k$-th nearest neighbor is less than or equal to $r$ is the probability that there are at least $k$ points within radius $r$:

$$F(r) = \mathbb{P}\{N(r) \geq k\} = 1 - \mathbb{P}\{N(r) \leq k - 1\}$$

Since $N(r)$ is Poisson-distributed:

$$F(r) = 1 - \exp(-\rho\pi r^2) \sum_{n=0}^{k-1} \frac{(\rho\pi r^2)^n}{n!}$$

Differentiate $F(r)$ with respect to $r$:

$$f(r) = \frac{dF(r)}{dr}$$
$$= \frac{d}{dr}\left(1 - \exp(-\rho\pi r^2)\sum_{n=0}^{k-1}\frac{(\rho\pi r^2)^n}{n!}\right)$$

Simplify the derivative:

$$f(r) = \left[2\rho\pi r \exp(-\rho\pi r^2)\sum_{n=0}^{k-1}\frac{(\rho\pi r^2)^n}{n!}\right] - \left[\exp(-\rho\pi r^2)\sum_{n=0}^{k-1}\frac{d}{dr}\left((\rho\pi r^2)^n\right)\frac{1}{n!}\right]$$
$$= 2\rho\pi r\exp(-\rho\pi r^2)\frac{(\rho\pi r^2)^{k-1}}{(k-1)!}$$
$$= \frac{2(\rho\pi)^k r^{2k-1}}{(k-1)!}\exp(-\rho\pi r^2)$$

This distribution is a generalized form of the exponential distribution in two dimensions. This derivation is an extension of the *nearest neighbor distribution* (Torquato et al., 1990) from considering the closest neighbor to considering the $k^{th}$ nearest neighbor.

The parameter $\rho$ in the nearest neighbor distribution prior for calculating the structure loss are determined through least-squares fitting. The results are shown in Fig. 6.

## A.2    IMPLEMENTATION DETAILS

**Hyperparameters**    We optmize a maximum of $1,000$ steps with an Adam (Kingma & Ba, 2015) optimizer. The optimization is stopped once the classifier decision is flipped. For our *Diffusion (Diff)* score distillation method, we employ different learning rates in the global and local latent space. Specifically, we use $\text{lr}_{\text{global}} = 0.001$ and $\text{lr}_{\text{local}} = 0.005$. Please note that the score distillation loss $L_{\text{sds}}$ is not affected by learning rate. In other words, the learning rate for this loss is always 1. For the *Autoencoder (AE)* method and *Adv. Attack (AA)* baseline, we use a learning rate of $0.01$. The weights of the losses are specified in Tab. 3a. These weights are empirically determined using a grid-search approach.

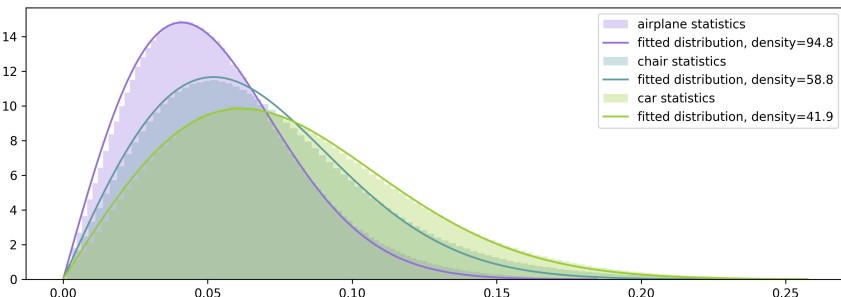

Figure 6: **Nearest Neighbor Distribution Fitting**. The histogram of nearest-neighbor distances aligns with the fitted probability density functions

|  | $\lambda_{sds}$ | $\lambda_{prox}$ | $\lambda_{st}$ | $\lambda_{div}$ |
|---|---|---|---|---|
| Adv. Attack (AA) | - | 0 | 0 | 0 |
| Adv. Attack + (AA+) | - | 0.1 | 0.3 | 0.03 |
| AutoEncoder (AE) | - | 0.1 | 0.3 | 0.03 |
| Diffusion (Diff) | 0.03 | 0.1 | 0.3 | 0.03 |

(a) **Hyperparameters** of our counterfactual explanation pipelines.

| Data | Accuracy |
|---|---|
| Chair | 1.00 |
| Airplane | 0.998 |

(b) **Accuracy** of our binary classifiers trained on ShapeNetCore.

**Pre-trained Generators** Due to the limitations of the architecture and the generative model provided by (Zamorski et al., 2019) that it is, *AE* is trained on the same ShapeNet category $c$ the classifier has been trained on. *Diff* is trained on the 55 categories of ShapeNetCore.

**Classifiers performance** We report in Table 3b the accuracy of our classifier on the binary classification task.

