# OpenReview forum: "Counterfactual Explanations for 3D Point-Cloud Classifiers"
_ICLR.cc/2025/Conference — Submitted to ICLR 2025_

### Official Review · Reviewer_Ke6E · 2024-10-25

**Soundness:** 3
**Presentation:** 4
**Contribution:** 3
**Rating:** 6
**Confidence:** 3

**Summary:**

The paper borrowed the idea of counterfactual explanations originally developed for 2D domain into 3D classification task. The idea is to generate samples that are close to an original sample but deceive the classifier. Differently from adversarial attacks, the generated samples should hold semantic meaning that may help better understand the root cause of misclassified samples.

**Strengths:**

1) The paper is well written and easy to follow.
2) It is self-contained and thoroughly explained even for the general audience which might not be familiar with the concept of counterfactual explanations.
3) The visualizations are aesthetically pleasing, which is important for explainability method to provide.

**Weaknesses:**

1) The method is compared only with DGCNN as a backbone. This architecture is indeed basic and elementary in point cloud classification, but it is not cover a wide range of possible architectures. It is recommended to select diverse architectures (PCT for transformer, maybe PointNet for simple learnable architecture, or more up-to-date networks like Recon).
2) The quantitative and qualitative analysis is mainly focused on the comparison between AE to diffusion, whereas the more profound idea here is the analysis of the results themselves. What can I learn from the visualizations about which elements are crucial for detecting a chair for example. I can see that the handle is transformed, but is it consistent across a wide range of chairs? is it the most prominent feature? (what if it is absent?)
3) Comparison to Other 3D XAI Methods: The paper does not provide a comparison with other XAI methods for 3D data. As a result, it is unclear how this approach performs relative to existing methods. A common strategy for XAI comparison is the perturbation test, which has also been applied in 3D contexts (e.g., see https://arxiv.org/pdf/2403.07706). While this comparison approach is particularly suitable for methods that rank points by importance, potentially differing from your approach, I believe an evaluation that allows for some level of comparison would be beneficial.

**Questions:**

1) It is difficult to detect what is the reason that the counterfactual example is deceive the classifier. For example, the elongated airplane (Fig. 3a) could cause the classifier misclassify the sample as a rocket for example, but maybe it will appear otherwise if it deceive the classifier to think that the example is a chair. Therefore I think: a) The division for a chair and not a chair (contains of many classes) is a bit vague. Perhaps it would be better to divide into a more perceivable groups (i.e. airplane against desk). b) The information of what is the misclassified class could have promote understanding.
2) In the qualitative results for airplanes, I observe three distinct counterfactual examples, each emphasizing different parts of the object (the wings, nose, and tail). How could your approach be applied to generalize across the entire airplane class? For instance, if I have a limited budget of points to remove or add for a new, previously unseen airplane, where should these resources be focused according to your method?
Minor edits:
1) Subsection header 3.1 should be ENCODER-DECODER PIPELINE (instead of ENCODE-DECODER PIPELINE - missing R)
2) in line 265 missing ) - $f(x) \neq f(\tilde{x})$ instead of $f(x) \neq f(\tilde{x}$.
3) line 523: "has been evaluated on a for a limited amount" should be corrected.

Overall, I think that the idea is good, the level of writing is appropriate and the paper is fluent. I decided to give the paper a positive rank, I believe that compare on top of more backbones, and against available XAI methods would make the paper more complete and I would have rank it even higher.

---

### Official Review · Reviewer_JV7z · 2024-10-31

**Soundness:** 2
**Presentation:** 2
**Contribution:** 2
**Rating:** 5
**Confidence:** 5

**Summary:**

The paper presents a new approach to generating counterfactual explanations (CEs) for 3D point-cloud classifiers, primarily targeting applications where transparency in neural network decisions is essential. The authors propose a diffusion-based CE generation method, specifically designed to retain semantic integrity while minimizing distortions to the original data.

**Strengths:**

1. The authors chose a meaningful topic with broad potential. Generating counterfactual samples doesn’t just benefit explainable AI; in today’s diffusion era, this could be instrumental in studying the domain gap between synthetic and real samples, offering promising avenues for further research. Great choice of focus.
2. The paper is well-written and easy to follow. It’s clear the authors put effort into making complex concepts digestible, which makes the research more accessible to a wide range of readers.

**Weaknesses:**

1. I feel the technique innovation is limited. I do not see specific design that is tailored for 3D point cloud; it just follows the diffusion based counterfactual explanation techniques in 2D field. The authors state that "Our work therefore diverges from existing CE methods by not focusing on 2D image-based models but embracing 3D data processing challenges, with a particular emphasis on point cloud classifiers." In my view, here is just a direct reuse of previous methods for 3D point cloud setting. Moreover, specific references should be given for "existing CE methods".
2. What is the difference between 2D image and 3D point cloud, in terms of counterfactual explanation?
3. While generating counterfactual explanations in this way is innovative, diffusion models have already been widely applied in 3D point cloud generation across various conditions and representations. I didn’t see any particularly clever or tailored use of the diffusion model that sets this method apart for counterfactual generation. It would be helpful if the authors could highlight any unique contributions of this method specifically for the counterfactual generation challenge.
4. About SDS loss. I’m not fully convinced about the need for SDS loss here. In the AIGC field, SDS is often used to bring in prior knowledge from diffusion, but it’s not entirely clear why it’s needed for counterfactuals in this case. While the results show that SDS improves generation quality, I’m not sure it’s contributing directly to the counterfactual sample generation itself. A bit more justification on this point would be useful, and perhaps some ablation experiments to assess whether it’s really essential for the counterfactual task.
5. The literature review is not comprehensive. Some relevant efforts are missing, such as Interpretable3D: An Ad-Hoc Interpretable Classifier for 3D Point Clouds, AAAI.
6. The paper could use more details on the structure of the diffusion model and the training setup. Information like the specific architecture, data size, optimizer settings, etc., would improve reproducibility and help readers understand exactly how the method was implemented. Adding these details would make the paper feel more complete.
7. For point cloud tasks, I think the autonomous driving setting is more important for explanation as it is safety-aware. But it seems that the proposed technique cannot be used for explaining 3D autonomous driving models.
8. The proposed method seems limited to specific types of classifiers, like DGCNN. Extending the approach to a broader range of 3D models would provide a more comprehensive evaluation.
9.  Considering the claim of “… to improve the memory footprint and the computational costs of diffusion-based CE generation,” the computational costs should be discussed. Diffusion models are computationally expensive, especially in high-dimensional data like 3D point clouds. The paper does not address strategies for managing computational load, which might limit practical applications.
10.The method is evaluated primarily on the ShapeNet dataset, which may not represent all 3D data scenarios. Evaluation on diverse datasets, especially those with more complex and noisy real-world data, would strengthen the validation.
11. The generated counterfactual explanations might be challenging for non-expert users to interpret without additional visualization tools or interpretive guidance.
12. A comparison with other explainable AI methods, even those adapted from 2D to 3D data, could offer insight into the specific advantages or trade-offs of the proposed approach.
13. Inconsistent Reference Formatting: For example, in L572 and L574, “ … Advances in Neural Information Processing Systems, ...” should be “ … In Advances in Neural Information Processing Systems (NeurIPS), …”; in L578, “ … International Conference on Machine Learning ...” should be “... In International Conference on Machine Learning (ICML), …”.
14.Typos: For example, in L265, “f(x” should be “f(x)”; punctuation should be added after equations in the Appendix.
15. Limitations and societal impact should also be discussed to provide a comprehensive understanding of the approach's practical constraints and broader implications. Addressing these aspects helps identify areas for improvement and promotes Explainable AI (XAI) development by anticipating potential ethical and societal challenges.

**Questions:**

Please see the above.

---

### Official Review · Reviewer_ovMT · 2024-11-01

**Soundness:** 2
**Presentation:** 3
**Contribution:** 3
**Rating:** 3
**Confidence:** 2

**Summary:**

This paper presents a diffusion-based method designed to generate counterfactual explanations (CEs) for 3D point cloud classifiers, aiming to fill a gap in explainable AI for 3D data. The authors introduce tailored loss functions and evaluation metrics to ensure semantic consistency and data fidelity, establishing a potential benchmark for future studies.

**Strengths:**

- The paper introduces a unique application of counterfactual explanations within 3D point cloud classification, offering a new approach to explainable AI in this area.
- By utilizing a diffusion model for CE generation, the method combines novel loss functions and constraints that enhance data fidelity and semantic consistency in 3D counterfactual examples.

**Weaknesses:**

- Although the paper claims to generate human-interpretable counterfactuals, it lacks a thorough quantification of the "semantic relevance" of these explanations, limiting the interpretability and effectiveness of the proposed approach.
- Experimental validation is limited to a single dataset and task (binary classification using ShapeNet). This narrow scope restricts the generalizability and fails to demonstrate the method's robustness across a broader set of data and classification tasks.
- The evaluation is restricted to a single point cloud classifier (DGCNN), without extending the analysis to other architectures. This narrow focus raises concerns about the method’s applicability across diverse model types.
- Although an ablation study on loss functions is provided, the paper lacks a detailed examination of each loss component's impact on the generated counterfactuals, which would strengthen claims regarding the method’s effectiveness.

Overall, the experimental results fall short of validating the authors' claims. While the idea is interesting, the findings remain unconvincing due to limited and narrowly focused evaluations.

**Questions:**

See weaknesses.

---

### Official Review · Reviewer_q5mx · 2024-11-04

**Soundness:** 3
**Presentation:** 3
**Contribution:** 3
**Rating:** 6
**Confidence:** 3

**Summary:**

This paper aims to generate 3D counterfactuals to investigate the performance of 3d point cloud-based network models. It studies CEs in 3D and 2D classifiers and contributes evaluation metrics designed to measure 3D CEs. They also contribute a benchmark for 3D CE point cloud classifiers. Specifically they use diffusion models to create 3D CEs with a counterfactual loss, to deceive the classifier, proximity and distribution losses to keep point cloud typical appearances and an diversity loss to generate a range of counterfactual explanations.

**Strengths:**

The method of learning an residual in an encoder-decoder pipeline to generate counterfactual samples seems straightforward and yet powerful.

The implementation is described in great detail making this work probably easily reproducible.

The used evaluation metrics are suitable and well motivated.

A lot of qualitative examples are given and, even if I do not agree with all their interpretations, they illustrate the performance of the method well.

**Weaknesses:**

The final loss is a complex combination of 4 losses. The ablation study does not seem to give a clear answer to see which losses are needed and what their individual contribution is. If I interpret the column correctly each loss is combined individually with L_cf, what makes sense because that is the counterfactual loss. However, combinations such as L_cf + L_prox + L_div but without L_st is unknown. But it is not only the loss combination, also the mixed performance does not give a high degree of confidence that all losses are needed or well balanced.

Given that this is, as the authors claim, the first generation of it's kind, it is hard to compare it to existing state of the art.

The qualitative examples could be a matter of debate. The concrete semantical changes in the airplane category could also stem from noise. Semantic changes of chairs seem also minor.

**Questions:**

In figure 3. the semantic change for b) is what exactly? The body is bent but that does not necessarily mean the sematic category is different? It would be helpful to have the idea of the authors in the figure caption instead of only , as in this case, in 4.2.2.

There are some works in the medical domain which do generate counterfactuals for 3D data but not pointcould data. I did not take this into account during my review to diminish my rating or to suggest this is not the first work but it would be helpful if the authors could position themselves to these works. Note that these are partially very early stage research articles:

Rasal, Rajat R., et al. "A Framework for Generating 3D Shape Counterfactuals." Geometric Deep Learning in Medical Image Analysis (Extended abstracts). 2022.

Maser, Michael, et al. "MoleCLUEs: Molecular Conformers Maximally In-Distribution for Predictive Models." NeurIPS 2023 AI for Science Workshop.

Peng, Wei, et al. "Latent 3D Brain MRI Counterfactual." arXiv preprint arXiv:2409.05585 (2024).

Ohara, Erik Y., et al. "MACAW 3D: A masked causal normalizing flow method for counterfactual 3D brain image generation." Medical Imaging 2024: Imaging Informatics for Healthcare, Research, and Applications. Vol. 12931. SPIE, 2024.

---

### Meta-Review · Area_Chair_7EbE · 2024-12-24

**Metareview:**

The paper proposes a novel method for generating counterfactual explanations for 3D point cloud classifiers using diffusion models. It introduces new loss functions and evaluation metrics tailored for 3D data and provides a benchmark to assess the quality of such explanations. The work seeks to fill a gap in explainable AI by moving from 2D image-based models to 3D data while maintaining semantic consistency and data fidelity.

Reviewers appreciated the focus on 3D explainability and acknowledged the potential value of the proposed benchmark. However, significant concerns were raised about the limited scope of the experiments, as evaluations were restricted to a single dataset (ShapeNet) and classifier (DGCNN). The lack of comparisons with other 3D explainability methods and the insufficient ablation studies for the loss functions were also flagged. Additionally, doubts were raised about the interpretability of the counterfactuals and the methodological novelty, as the approach appeared to largely adapt existing techniques from 2D to 3D without substantial innovations. Unfortunately, the authors did not provide response during the rebuttal.

Considering the above factors, the AC recommends rejection. While the topic is interesting and has potential, the lack of response from the authors and unresolved critical issues prevent the paper from being accepted at this stage.

**Additional Comments On Reviewer Discussion:**

Please refer to the meta review.

---

### Decision · Program_Chairs · 2025-01-22

Reject